# Visual System Inspired Algorithm for Enhanced Visibility in Coronary Angiograms (VIAEVCA)

**DOI:** 10.3390/biomimetics10010018

**Published:** 2025-01-01

**Authors:** Hedva Spitzer, Yosef Shai Kashi, Morris Mosseri, Jacob Erel

**Affiliations:** 1School of Electrical Engineering, Iby and Aladar Fleischman Faculty of Engineering, Tel-Aviv University, Tel-Aviv 69978, Israel; shaikashi@gmail.com; 2Sackler Faculty of Medicine, Tel-Aviv University, Tel-Aviv 69978, Israel; 3Cardiology Department, Meir Medical Center, Kfar-Saba 95847, Israel; 4Nuclear Cardiology Unit and CCT Service, Meir Medical Center, Kfar-Saba 95847, Israel

**Keywords:** blood vessel enhancement, coronary angiography, lateral interaction algorithm

## Abstract

Numerous efforts have been invested in previous algorithms to expose and enhance blood vessel (BV) visibility derived from clinical coronary angiography (CAG) procedures, such as noise reduction, segmentation, and background subtraction. Yet, the visibility of the BVs and their luminal content, particularly the small ones, is still limited. We propose a novel visibility enhancement algorithm, whose main body is inspired by a line completion mechanism of the visual system, i.e., lateral interactions. It facilitates the enhancement of the BVs along with simultaneous noise reduction. In addition, we developed a specific algorithm component that allows better visibility of small BVs and the various CAG tools utilized during the procedure. It is accomplished by enhancing the BVs’ fine resolutions, located in the coarse resolutions at the BV zone. The visibility of the most significant clinical features during the CAG procedure was evaluated and qualitatively compared by the consensus of two cardiologists (MM and JE) to the algorithm’s results. These included the visibility of the whole frame, the coronary BVs as well as the small ones, the main obstructive lesions within the BVs, and the various angiography interventional tools utilized during the procedure. The algorithm succeeded in producing better visibility of all these features, even under low-contrast or low-radiation conditions. Despite its major advantages, the algorithm also caused the appearance of disturbing vertebral and bony artifacts, which could somewhat lower diagnostic accuracy. Yet, viewing the processed images from multiple angles and not just from a single one and evaluating the cine mode usually overcomes this drawback. Thus, our novel algorithm potentially leads to a better clinical diagnosis, improved procedural capabilities, and a successful outcome.

## 1. Introduction

Coronary artery disease (CAD) is a major global cause of death [1], caused by atherosclerotic plaque buildup. It may cause gradual narrowing of the inner side of arteries over time or abrupt rupture and clotting that end up with partial or total blockage of the blood flow. The current gold standard for the detection and invasive treatment of CAD is conventional X-ray CAG [2]. A contrast agent is injected during the acquisition of CAG to improve the visibility of the BV and their blood flow. A CAG unavoidably exposes the patients to a limited controlled dose of harmful X-ray radiation and toxic contrast agents [3].

The capabilities to enhance CAG visibility post-acquisition are based on computational methods that modify brightness, contrast, and background by various algorithms. The quality of CAG images still suffers from low contrast (mainly due to dye clearance and low-radiation fluoroscopy runs), non-uniform illumination issues, and the presence of other body organs, which makes the diagnosis and medical interventional procedures a challenging task. A comprehensive algorithm that can simultaneously enhance the visibility of all image components, without losing details and even revealing important clinical information, is therefore, clearly needed.

Improved CAG visibility has the potential to help also in selecting the required equipment for the catheterization procedure. It supports smooth and faster navigation with the various catheters, especially with the thin wire inserted into the coronary blood vessel (CBV), which is commonly used during invasive treatment. This allows accurate deployment of balloons and stents over it, precisely at the right locations. A successful procedure includes many additional tools and elements, which are all dependent on visibility, such as placing the tip of the catheter at the proper angle in the orifice of the vessel, avoiding damage to the delicate endothelial tissue when the dye is injected at high pressure, or when entering and navigating with the tip of the wire inside the vessel and later deploying a stent over it. An additional advantage is avoiding unnecessary invasive treatment when identifying good flow through collaterals that bypass the obstructive lesion or that originate from other CBVs and supply sufficient blood to the same viable myocardial territory. Previous algorithms did not relate to this type of CAG enhancement and visibility.

Over the years, numerous studies have been developed for the segmentation and enhancement of BVs applied to different anatomical areas and imaging modalities [4,5]. These classical algorithms dealt mainly with these challenges through second derivative algorithms (i.e., Hessian algorithms) [6] and other algorithms that tried to improve the dynamic range of the image in order to solve illumination problems, such as contrast limited adaptive histogram equalization (CLAHE) [7,8], And additional study suggested morphological filters [9].

Hessian-based methods have commonly been applied for the purposes of blood vessel segmentation and enhancement in CAG images [10,11,12,13]. Although these studies obtained good segmentation of the main coronary arteries, only a small repertoire of the successful segmentation of small BVs was obtained. Moreover, these Hessian derivative algorithms also caused enhanced noise in the image [14]. Truc and his colleagues (2009) tried to reduce the noise by using a Hessian algorithm and applying wedge filters, as preprocessing components.

Some of these publications added additional factors that enhance and improve the blood vessel’s appearance, such as adding the G-L differential kernel [15], which yielded better results than, for example, the classical CLAHE algorithm. Results of CLAHE algorithms have appeared many times with exaggerated results and with additional noise [16].

Few computational algorithms have been developed to reduce the amount of toxic X-ray radiation and contrast agent dose while minimizing image quality degradation [1,17,18]. For enhancing the visibility at low doses, different methods have been used, such as decreasing the noise through principal component analysis (PCA) [17], adding the isolated vessel layer to the low-dose image [18], and using a convolutional neural network (CNN) algorithm [1]. Exposure of additional BVs under the condition of low contrast was shown by using fractional differential kernels [19] and additional classical methods, such as Hessian-based methods [15] or lateral inhibition-based models.

Due to the substantial research efforts dedicated to improving CAG image quality issues at standard radiation, our study mainly refers to the most recent and relevant studies. Those studies have been developed in order to overcome several image visibility drawbacks, such as non-uniform illumination, artifacts and noise, low contrast, and interference of vertebral objects [15,18,20,21].

A group of other recent studies challenged the efficient segmentation of BVs through background subtraction algorithms. These studies aimed to isolate the BVs, in still or in video images, from the bony “artifacts” of the spinal cord and additional organs. The purpose of this approach is to allow the observation of BVs without “disturbing” vertebral bodies, such as bones and the spinal cord. These studies computationally separate the CAG image into two [20,22] and sometimes three different layers [18,21], namely, vessels, breathing, and background (spine and hard tissues) layers. Since the spatial information by itself cannot separate efficiently the three layers, most of the subtraction operations used mutual information of spatial and temporal aspects. Most of these algorithms utilized the known Robust Principal Component Analysis (RPCA) method [18,20,21]. Such a method enables the separation of the image into a matrix that is decomposed into a low-rank and sparse matrix, which represents the background and vessel layers. These subtraction studies [18,20,21] succeeded in separating the background from the BV image, but none of them included the small BVs in their BV layer.

In recent years, CNNs have become a common method for coronary blood vessel segmentation [8,23,24,25,26]. Most of the studies on this topic did not include exposure of small blood vessels in their ground-truth image [8,23,26], and therefore, their segmentation results did not reveal exposure of small blood vessels. Only more recent papers, from 2021 [24,25], presented ground-truth images that supplied a larger repertoire of blood vessels, including the small blood vessels, to the U-net and Random Forests Classifier algorithms. Their results yielded better performance than the previous algorithms, but still, the visibility of small blood vessels was limited [25].

Although many studies and computational methods addressed the CAG image quality issues, it seems that solutions for the clear appearance of small blood vessels and visibility of the content inside the large blood vessels with their relevant catheterization apparatus are still required. Revealing these details and objects can be critical in allowing an efficient diagnosis. We present here a single algorithm that addresses both the visibility of BVs and the catheterization tools. It simultaneously reduces noise and illumination artifacts and can also enhance the visibility of BVs under the conditions of a low-contrast agent dose and reduced radiation. The algorithm is based on a new biologically inspired mechanism and a computational model (lateral facilitation) [27,28].

## 2. Proposed Algorithm

### 2.1. Rational of the Algorithm

In this study, we developed and applied different algorithm components for enhancing the visibility of CBVs and the various tools utilized during the CAG procedure. For this purpose, we first used a preliminary stage aimed at enhancing the visibility of the relevant region of interest (ROI) in the images, namely, the CBVs. For a successful diagnosis, cardiologists need to properly evaluate the whole CBV system, including the small vessels, and identify all significant obstructive lesions. Enhancing the visibility of the catheterization tools within the blood vessels is also important for successful invasive procedures. The algorithm was developed to overcome quantum noise, artifacts, non-uniform illuminations, and low contrast due to a low-radiation and/or low-contrast agent dose, which can all substantially decrease visibility.

The suggested algorithm is based on Multiscale Gabor filters having different orientations and scales (similar to what occurred with the different BVs). The main contribution of the algorithm to the BV enhancement is inspired by a visual system mechanism and its model [27]. This mechanism is known as lateral facilitation or lateral interactions, while the mechanism of contour integration is also included in the suggested model [27,28,29]. (2) The model suggests an additive facilitation sub-threshold signal, which is added to an area outside the classical receptive field but that is induced at the optimal relevant orientation of the classical receptive field. This sub-threshold response may be enhanced to become suprathreshold by the additive signal of the neighboring receptive fields (RFs) (in the case that a similar orientation preference is shared between the two adjacent receptive fields). The possibility that the additive signal will cause a super-threshold response and fulfill contour integration can be regarded as determined by an educated guess signal (additive signal). The line and texture completion of the blood vessels enables us to enhance their structure at the expense of part of the noise in an image.

### 2.2. Overview

Our algorithm for blood vessel visibility enhancement is based on several building blocks, which are summarized in the schematic diagram of Figure 1. Each component is represented by a different background color.

The algorithm’s first stage (Figure 1A, green block) performs Gabor filtering in several orientations and scales, which is similar to what is performed with the simple cells’ receptive fields of the visual system. The second stage (Figure 1B, pink block) performs the line completion algorithm, which is based on computational models aimed to supply mechanisms and models to visual phenomena, such as lateral facilitation, and collinear interaction stages. The third stage (Figure 1C yellow block) performs the extraction of the blood vessel’s region of interest (ROI). Then, in the fourth stage (Figure 1D cyan block), the ROI result is used to extract the blood vessels from the image of the line completion results (Figure 1E pink block). The resulting image contains the blood vessel edges’ texture (which has been obtained from the line completion) but only at the zone of the blood vessel locations (BV ROI). The purple block represents blood vessel visibility enhancement after the integration of the blood vessel edge texture with the original image (DC of the image). The algorithm results (purple block) have been presented to the cardiologists for estimation and feedback. The additional stage of algorithm corrections, according to the cardiologists’ feedback, is the final estimations, which are presented in the Section 5.

### 2.3. Orientation and Scale Filtering (Gabor)

The first stage of the algorithm is intended to extract edges, which are built from different scales (j) and different orientations (θ). The edge image, $s\left(x,y,j,\theta \right)$, is obtained through the convolution of Gabor kernels$\;G_{\hat{\theta }}\left(x,y\right)$, Equation (2), while using a Gaussian Image Pyramid deconstruction of the input image $I\left(x,y\right),\;\mathrm{which}\;\mathrm{is}\;\mathrm{built}$ from different scale images, $I^{j}\left(x,y\right)$. The edge image, thus, is built of edges with different orientations and different spatial resolutions. Equation (1) represents the calculation of the Gabor kernels at different orientations:(1)$$G_{\hat{\theta }}\left(x,y\right)=\frac{1}{g}e^{-\frac{{\left(x-x_{0}\right)}^{2}+{\left(y-y_{0}\right)}^{2}}{\sigma^{2}}}\cdot \cos\left(\frac{2\pi }{\lambda }\cdot \left(x-x_{0}\right)\cdot \cos\left(\hat{\theta }\right)+\left(y-y_{0}\right)\cdot \sin\left(\hat{\theta }\right)\right)$$
where the frequency ($\frac{1}{\lambda }$) and the orientation (θ) of the spatial filter are determined by the frequency and orientation at each spatial location $\left(x,y\right)$. σ is the standard deviation of the Gaussian envelope.$\;\left(x_{0},y_{0}\right)$ is the spatial location of the center of the Gabor filter. g is chosen as the normalization factor.

To estimate the orientation feature of images’ edges (stimulus), we first convolved the Gabor kernel at different orientation values:(2)$$s\left(x,y,j,\theta \right)=\left\{\begin{array}{l} \underset{n}{\sum }\underset{m}{\sum }I^{j}\left(m,n\right)\cdot G_{\theta }\left(x-m,y-n\right)\;\;\;\;\;\;x,y\;at\;RFs\;center\\ \;\;\;\;\;0\;\;\;\;\;\;\;\;\;\;\;\;\;\;\;\;\;\;\;\;\;\;\;\;\;\;\;\;\;\;\;\;\;\;\;\;\;\;\;\;\;\;\;\;\;\;\;\;\;\;\;\;\;\;\;O.W \end{array}\right.$$
where $I^{j}\left(x,y\right)$ is a Gaussian Image Pyramid deconstruction of $I\left(x,y\right)$, and j represents the spatial scale.

After the edge (sparse) image $s\left(x,y,j,\theta \right)$ is obtained, we determine the optimal edge orientation at each location (*x*, *y*), separately for each scale (j). Therefore, in Equation (3) $S\left(x,y,j\right),\;$Figure 2C is chosen as the strongest convolution response (Equation (2)).
(3)$$S\left(x,y,j\right)=\arg\underset{\theta \in \left[0,90\right]}{\max}\left(s\left(x,y,j,\theta \right)\right)$$

Due to the Gabor structure, which contains positive and negative areas, the optimal orientation edge image $S\left(x,y,j\right)$ contains both the strongest response in the positive and the negative value ranges. These two maximum responses are considered separately, as shown in Equations (4) and (5).
(4)$$c_{P}\left(x,y,j\right)=\max\left(S\left(x,y,j\right)\;,0\right)$$
(5)$$c_{N}\left(x,y,j\right)=\max\left(-S\left(x,y,j\right)\;,0\right)\;$$where $c_{P}$ and $c_{N}$ represent the largest positive and negative responses, respectively.

For the sake of simplicity, from this point and further on, all the equations refer to P and N values as one group, although the computations are independent. The positive and negative components, until the components are separated, are recombined at the final stage of the algorithm. $c\left(x,y,j\right)$ is, therefore, used to represent both $c_{P}\left(x,y,j\right)$ and $c_{N}\left(x,y,j\right)$.

To decrease the amount of noise in the image, all intensity responses below 5% are considered here as noise (Equation (6)). The threshold, thr, is consequently calculated as the 5% percentile, across all the maxima response values and across all the spatial locations in the edge image $c\left(x,y,j\right)$. Equation (6) is shown below.
(6)$$c\left(x,y,j\right)=(c\left(x,y,j\right)>thr)\;\cdot c\left(x,y,j\right)\;$$
where
(7)$$thr=\pm \max\left|c\left(x,y,j\right)\right|\cdot 0.95$$

Following the lateral facilitation model [27], we applied the Naka–Rushton Equation $R_{RF}{\left(x,y\right)}_{j}$ (Equation (8)), Figure 2D, which was suggested as a response to the V1 simple cell’s responses to the edge images $c\left(x,y,j\right).$ This operation enables us to enhance the gain response in reference to a specific stimulus intensity range.
(8)$$R_{RF}{\left(c\left(x,y,j\right)\right)}_{j}=\frac{c{\left(x,y,j\right)}^{n}}{c{\left(x,y,j\right)}^{n}+\sigma_{NR}^{n}}$$
where $R_{RF}{\left(x,y\right)}_{j}$ is the Naka–Rushton response of a cRF cell at location (*x*, *y*) and scale *j*. For the sake of simplicity, the NR parameters *n* and $\sigma_{NR}$ are considered as constants.

### 2.4. Line Completion Algorithm

Figure 3 shows the schematic diagram of the line completion algorithm’s component.

## 3. Lateral Facilitation

### 3.1. Additive Signal

Following the rationale of the algorithm (Section 2.1), the algorithm is presented in terms of the model of line completion, i.e., lateral facilitation [27]. An additive signal (AdS) is added beyond the response to the classically relevant receptive field (*RF*), as an educated guess response (Equation (15)). It is taken, therefore, as a sub-threshold response that is added to the *RF* optimal response, which has been yielded from the preferred appropriate orientation receptive field. Consequently, this response is calculated as a sub-threshold signal, which is spatially located along its relevant *RF*. It becomes suprathreshold only when there is an “agreement” with the neuronal mechanism of lateral interaction [28,29]. For the sake of simplicity, we present first the additive signal spatial decay profile, which is built from Gaussian functions (Equations (9)–(12)), and then the induced additive signal response, while also including its intensity profile and not only its spatial profile (Equation (13)); see Figure 2C.

Since the additive signal, AdS (Equation (9)), is added collinearly, it is expressed as a Gaussian mask located along the *RF* orientation, *x*-axis, which indicates the collinearity to the core of the receptive field. This additive signal, Ads, is added to the maximum responses of orientation and resolutions, $s\left(x,y\right)$, as sub-threshold signals.
(9)$$AdS\left(x_{i},y_{i},\hat{\theta }\;\right)=\frac{1}{2\sigma_{x}}e^{\left(-\frac{1}{2}\left(\frac{x^{\prime }}{\sigma_{x}}\right)\right)}$$
where $x^{\prime }$ is the collinear axis,$\;\hat{\theta }$ is the orientation of the optimal receptive field, $\mathrm{and}\;\sigma_{x}$ is defined as the designated number of pixels needed for line completion along the θ orientation.

To enable flexibility in the connectivity between collinearity additive signals, which is supposed to perform an efficient line completion, the model suggests a second axis Gaussian, $y^{\prime }$, (perpendicular to the *RF* length—$x^{\prime }$ axis), as a $\mathrm{weighted}\;\mathrm{function}\;\left(\mathrm{WF}\right)\;$(Equation (11)). This component enables additivity between a neighbor’s *RF* with also only similar orientations.
(10)$$\mathrm{WF}\left(x_{i},y_{i},\hat{\theta }\;\right)=\frac{1}{2\sigma_{y}}e^{\left(-\frac{1}{2}\left(\frac{y^{\prime }}{\sigma_{y}}\right)\right)}$$
where $\sigma_{y}$ is the weight function of the decay of the degree of the additive signal collinearity.

The final additive signal profile ($AdS_{f}$) is presented by a two-dimensional Gaussian function (Equation (12)).
(11)$$AdS_{f}\left(x_{i},y_{i},\hat{\theta }\;\right)=AdS\left(x_{i},y_{i},\hat{\theta }\;\right)\times \mathrm{WF}\left(x_{i},y_{i},\hat{\theta }\;\right)=\frac{1}{2\sigma_{x}\sigma_{y}}e^{\left(-\frac{1}{2}\left(\frac{x^{\prime }}{\sigma_{x}}+\frac{y^{\prime }}{\sigma_{y}}\right)\right)}$$
where $x^{\prime }=x_{i}\cos\left(\hat{\theta }\right)+y_{i}sin\left(\hat{\theta }\right),\;y^{\prime }=y_{i}\cos\left(\hat{\theta }\right)-x_{i}sin\left(\hat{\theta }\right)$.

The additive signal, which includes its intensity and spatial profile is presented as follows:(12)$$R{F_{AdS}}_{\theta ,j}\left(x_{i},y_{i}\right)=f\left(d^{i},RF^{i}{\left(x_{i},y_{i}\right)}_{\theta ,j}\right)=\left\{\begin{array}{l} a\cdot R_{RF}^{i}{\left(c(x_{i},y_{i})\right)}_{j,\theta }\times AdS_{f}\left(x_{i},y_{i},\theta \right)\;\;\;\;\;\;\;d^{i}>r_{RF}\\ 0\;\;\;\;\;\;\;\;\;\;\;\;\;\;\;\;\;\;\;\;\;\;\;\;\;\;\;\;\;\;\;\;\;\;\;\;\;\;\;\;\;\;\;\;\;\;\;\;\;\;\;\;\;\;\;\;\;\;\;\;\;\;\;\;d^{i}\leq r_{RF} \end{array}\right.$$
where $R_{RF}^{i}{\left(c(x_{i},y_{i})\right)}_{j,\theta }$ is the cRF response of the inducing *RF*. $d^{i}$ is the distance from the center of the inducing RF to the edge of the receptive field and the distance of the additive signal, which includes the regions outside of the classical receptive field in the preferred orientation. $r_{RF}$ is the distance between the center of the receptive field to its edges. a represents the strength of the induced *LF* (lateral facilitation).

### 3.2. Collinear Interactions

This stage of the model examines the possibility of the lateral facilitation operation at each image location; see Figure 2C. In other words, the model examines whether the sub-threshold additive signal, $R{F_{AdS}}_{\theta ,j}\left(x_{i},y_{i}\right)$ (Equation (13)), at each specific *RF*, is changed to be super-threshold, in dependence on the responses at the neighborhood of the collinear neighborhood regions, $R{F_{AdS}^{k\left(x_{i},y_{i}\right)}}_{\theta ,j}\left(x_{i+\cdot },y_{i+\cdot }\right)$. The model, therefore, examines whether there are collinear matching flankers that can overlap with their additive signals and, consequently, cause a super-threshold response. This additive signal interaction, thus, can fulfill the condition for the lateral facilitation operation (Equation (14)).
(13)$$RF_{LF}{\left(x_{i+\Delta },y_{i+\Delta }\right)}_{\theta ,j}=R{F_{AdS}^{i}}_{\theta ,j}\left(x_{i},y_{i}\right)+R{F_{AdS}^{k}}_{\theta ,j}\left(x_{i+\Delta },y_{i+\Delta }\right)$$
where $RF_{AdS}$ is the receptive field additive signal at a specific location, while i and k are the indexes that present two different neighboring receptive fields, $RF_{AdS}$.

Until this stage, we considered only the facilitation obtained from the responses of adjacent collinear receptive fields, Equation (14). The model should also consider a possible response, $R_{RF}$, in case there is a stimulus located in the region of the overlapping summated receptive field with their additive signals, $RF_{LF}$. (The above consideration agrees with many experimental studies that have been completed on lateral facilitation and lateral integration (i.e., ref. [24])).

The lateral facilitation procedure, Equation (15), includes possible summation from all the receptive fields with their overlapping additive signals, along with the $R_{RF}$, located in the summation spatial area; see above.
(14)$$LF{\left(x,y\right)}_{\theta ,j}=R_{RF}\left(x,y\right)+\sum_{i}RF_{LF}^{i}{\left(x_{i+\Delta },y_{i+\Delta }\right)}_{\theta ,j}$$
where $LF{\left(x,y\right)}_{\theta ,j}$ represents the sum of the lateral facilitation signals at each *RF* location (*x*, *y*), along with $R_{\mathrm{RF}}\left(x,y\right)$; see Equation (9). Note that there is an additional stage of the threshold; see Equations (16) and (17).

The same logic of fulfilling the lateral facilitation is included for the whole image at all locations, through the summation presented in Equation (16). $LF{\left(x,y\right)}_{\theta ,j}$ presents the last stage of the lateral facilitation procedure, for the whole image, but before an additional stage of thresholding, as presented in Equations (17) and (18).
(15)$$LF{\left(x,y\right)}_{\theta ,j}=c_{\theta ,j}\left(x,y\right)+\sum_{i}RF_{LF}^{i}{\left(x_{i+\Delta },y_{i+\Delta }\right)}_{\theta ,j}$$
where $LF{\left(x,y\right)}_{\theta ,j}$ represents the sum of the lateral facilitation signals at each *RF* location (*x*, *y*), along a similar orientation θ and scale j. $c_{\theta ,j}\left(x,y\right)$ represents the cRF response at location (*x*, *y*).

Following the previous section that described lateral facilitation, $LF{\left(x,y\right)}_{\theta ,j}$, which is fulfilled only in the case that there is overlapping between adjacent additive signals of adjacent receptive fields, an additional threshold level, the $thr_{j}$ stage, has been added, $LF\_thr{\left(x,y\right)}_{\theta ,j}$; see Equation (17). The level of the threshold is determined according to percentile measure; see Equation (18). The aim of the thresholding stage, Equation (17), is to suppress *LF* responses, which are too low. This stage allows the algorithm a further reduction of the enhanced noises.
(16)$$LF\_thr{\left(x,y\right)}_{\theta ,j}=\max\left(0,\;LF{\left(x,y\right)}_{\theta ,j}\right)\cdot H\left(LF{\left(x,y\right)}_{\theta ,j}-thr_{j}\right)$$
where H is a soft-threshold function.
(17)$$thr_{j}=P_{\gamma_{j}\%}^{j}\left(LF{\left(x,y\right)}_{\theta ,j}\right)$$
where $P_{\gamma_{j}}^{j}$ is the $\gamma_{j}\%$ percentile of the lateral facilitation response values, $LF{\left(x,y\right)}_{\theta ,j}$, in Equation (18), across the whole image. $\gamma_{j}$ is the percentile constant per scale j.

The algorithm components have been calculated separately for each response feature: polarities (*P*, *N*), orientations (θ), and scales ($j)\;\left(\mathrm{see}\;\mathrm{above}\right)$. The following stages describe how the different components are combined to have final lateral facilitation responses while including all their features. The integration of the different components has been described separately for each algorithm’s feature.

Polarities: The simulations have been treated separately for the positive and negative responses (Section 2.3; Equations (4) and (5)). However, our model omits writing each component twice, for the sake of simplicity in the previous sections.

We first start with two *LF* polarities, the negative $LF\_thr{{\left(x,y\right)}^{N}}_{\theta ,j}$ and positive $LF\_thr{{\left(x,y\right)}^{P}}_{\theta ,j}$.
(18)$$LF\_thr{{\left(x,y\right)}^{O}}_{\theta ,j}=LF\_thr{{\left(x,y\right)}^{P}}_{\theta ,j}-LF\_thr{{\left(x,y\right)}^{N}}_{\theta ,j}$$

Orientations: In the following stage, the algorithm sums up all the responses of the optimal orientations, which have been found for each Gabor Orientation, $LF\_thr{{\left(x,y\right)}^{O}}_{\theta ,j}$ (Equation (2)).

During this summation process, we would like to also gain noise reduction, and therefore, we enhance the larger responses and suppress the smaller responses, by raising $LF\_thr{{\left(x,y\right)}^{o}}_{\theta ,j}$ over the power of an integer m1≥1. To maintain the response sign (positive or negative response) of $T_{j}$, the calculation is separated into sign and absolute value components, in the following manner:(19)$$T_{j}=\sum_{\theta }\left[{\left(\left|LF\_thr{{\left(x,y\right)}^{O}}_{\theta ,j}\right|\right)}^{m1}\cdot sign\left(LF\_thr{{\left(x,y\right)}^{O}}_{\theta ,j}\right)\right]$$

To maintain the original *LF* intensity values range, while preserving the values signs and the enhanced contrasts, the algorithm applies an inverse operation to Equation (20):(20)$${T_{j}}^{\prime }=sign\left(T_{j}\right)\cdot \sqrt[{m1}]{\left|T_{j}\right|}$$

Scales: Equations (22) and (23) describe the operation of combining the collapsed scales. This is completed similarly as is completed for the recombination of the different orientations (Equations (20) and (21)). However, in this case, we had to consider the different scale image sizes. Before the scale recombination, each ${T_{j}}^{\prime }$ (Equation (21)) is interpolated back into the size of the original image scale (see Section 4). The resized set of ${T_{j}}^{\prime }$ responses are notated as ${T_{j}}^{\prime \prime }$.
(21)$$T=\sum_{j}\left[{\left(\frac{1}{\tau_{j}}\left|{T_{j}}^{\prime \prime }\right|\right)}^{m2}\cdot \mathrm{sign}\left({T_{j}}^{\prime \prime }\right)\right]$$
where m2 is constant (m2≥1), and$\tau_{j}$ is a scale-dependent constant, which is defined per each scale.
(22)$$T_{E}=\gamma \cdot \mathrm{sign}\left(T\right)\cdot \sqrt[{m2}]{\left|T\right|}$$
where γ is a normalization constant.

### 3.3. BV ROI

The extraction stage of the region of interest (BV ROI) of the BV is designed to extract the locations of BVs, the catheterization apparatus, and the contrast agent (Figure 4). It is completed, thus, also to locate and enhance the catheterization apparatus, only at these spatial BV areas. The ROI estimate image is computed for each scale (j) separately, at the first stage (Figure 4A). The ROI estimate is a binary image, while the white pixels represent the ROI, and the black pixels represent the background. The binary image is calculated according to the compound’s signal intensity values ${T_{j}}^{\prime }\left(x,y\right)$ (Equation (23)), through the following stages. First, the histogram of each compound signal scale is computed, while each bin represents the probability of the range of intensity values of the compound responses $T_{j}\prime \left(x,y\right)$. The intensity threshold is calculated according to Otsu’s method. Otsu’s method chooses a threshold that minimizes the intra-class variance of the threshold black and white pixels, and therefore, provides a simple and compatible threshold measure that is appropriate for determining an estimate that is based on intensity responses. Equation (24) applies Otsu’s threshold to create the binary image.
(23)$$wb\;\left(x,y,j\right)=\left\{\begin{array}{c} 1\;\;\;if\;T_{j}^{\prime }\left(x,y\right)>\mu_{T}\left(j\right)\;\\ 0\;\;\;\;else \end{array}\right.$$
where $\mu_{T}\left(j\right)$ is the Otsu threshold found for $T_{j}^{\prime }$.

The white pixels ($wb\;\left(x,y,j\right)=1)$ represent the spatial locations (*x*, *y*) of the BV ROI at the estimated image (Equation (24)), at each scale (j).

The additional ROI stage (Figure 4B), Equations (25) and (26), enables smooth continuity of the ROI region while selecting the major components in the image. The rationale of this stage is based on preserving the continuity of the physical BV tree, which could have been disrupted during the acquisition of the 2D CAG. Additional factors can interfere with achieving a non-ideal video image, which is needed for an optimal diagnosis. These factors are as follows: 1. Those that originate from deficient image quality due to limited X-ray or contrast agent doses, quantum noise, screening angle, non-uniform illumination, obstructions by different body organs, etc. 2. Disruptions that could have occurred from the threshold holding stages, at an earlier algorithm stage (Equation (24)). This threshold stage generates merely an estimate of the ROI; therefore, it may lead to some ROI areas that do not cross the threshold. This can lead to irregularities in the BV’s structure, such as black holes or gaps in the binary images. Thus, a selection operation such as morphological closing (Equations (25) and (26)) seems appropriate. (It is worth noting that the usage of morphological closing on the binary image cannot create irrelevant or new information, since this image will only be used for ROI extraction).

The dilation operation of $wb\;\left(x,y,j\right)$ is obtained from the following:(24)$$wb_{dilated}\;\left(x,y,j\right)=\cup_{t\in B}wb{\left(x,y,j\right)}_{t}\;$$
where *B* is a binary structuring element and $wb{\left(x,y,j\right)}_{t}$ is the translation of wb by t. The closing operation of $wb\;\left(x,y,j\right)$ is obtained from the following:(25)$$wb_{closed}\;\left(x,y,j\right)=\cap_{t\in B}wb_{dilated}\;{\left(x,y,j\right)}_{-t}$$
where *B* is a binary structuring element with a radius $r_{j}$, $\mathrm{and}\;wb_{dilated}\;{\left(x,y,j\right)}_{-t}$ is the translation of $wb_{dilated}\;\left(x,y,j\right)$ by −t.

The two preceding stages, the ROI estimate creation (Equation (24); Figure 4A) and the morphological closing (Equation (26); Figure 4B1)), which yielded a binary continuous BV tree, enabled us to use the connected-component labeling (CCL) algorithm, separately for each scale. This capacity of the CCL (Figure 4B2) to enhance the ROI was enabled after the creation of the largest connected tree, $wb_{closed}\;\left(x,y,j\right)$ (Equations (24)–(26)). The final stage of the CCL algorithm accounts for only the largest components in the image, $wb_{closed}\;\left(x,y,j\right)$, which are represented by $wb_{cc}\;\left(x,y,j\right)$.

Up to this stage, each ROI candidate (Equations (24)–(26)) was considered separately for each image scale, whereas the final ROI must include all the scales. The final stage of the ROI BV algorithm (Figure 4C), consequently, provides weighs function information (Figure 4C1; Equations (27) and (28)) from the different binary image scales $wb_{cc}\;\left(x,y,j\right)$, to enable the integration of various BV scales in a later stage (Figure 4C2, Equation (29)).

The main purpose of this part of the algorithm, as mentioned above, is to enhance the information inside the large BV that might contain information on additional smaller BVs, or on the catheterization tools. To perform this enhancement, we reorganize the histogram of the scale j partitions into new scale groups k, through different sizes of the interval (bins) and their degree of overlapping (bins’ width in the histogram). This partition is completed by smoothing the median values of the intervals of the samples (bins) of the histogram sizes of the different BV scales (Methods 3.4). The scale medians *L* (Equations (27) and (28)) have been chosen as the interval width of the number of image scales that are included in this interval. While $k\left(i\right)$ is intended to determine the degree of overlap between the bins, it is expressed through the initiation of each propagated interval; see Equation (27). The information of the different binary images of the scale information is, therefore, expected to be reorganized into better new partitions, which will follow the flexible size of the BV along their different locations.
(26)$$\hat{wb_{k}}\left(x,y\right)=Med\left(wb_{cc}\;\left(x,y,k\left(i\right)\right),\;\;\;\;wb_{cc}\;\left(x,y,k\left(i\right)+1\right),\ldots wb_{cc}\;\left(x,y,k\left(i\right)+L\right)\right)$$
where L is the width of the median interval, and i=1:N−L.

The median is calculated as follows:(27)$$Med\;\left(X\right)=\left\{\begin{array}{c} X\left[\frac{L}{2}\right]\;if\;L\;is\;even\\ \frac{\left(X\left[\frac{L-1}{2}\right]+X\left[\frac{L+1}{2}\right]\right)}{2}\;if\;L\;is\;odd \end{array}\right.$$

X is defined as the ordered list of values in the dataset, and L is the number of values in the dataset.

After the model’s stage that suggested how to reorganize the different image scales, through the new intervals, $WB\left(x,y\right)$, Equation (29) shows how the different scale binary images $\hat{wb_{k}}\left(x,y\right)$ are summed into a single multiscale ROI image (Figure 4C2).
(28)$$WB\left(x,y\right)=\sum_{k=1}^{N}\hat{wb_{k}}\left(x,y\right)$$
where N is the number of $\hat{wb_{k}}\left(x,y\right)$ binary images.

Since $WB\left(x,y\right)$, Equation (28), reflects the summation of the values of the different scales from each location, this summation can obtain a response, which is beyond the value one at specific locations. This is derived from obtaining agreement of the same value from more than one scale. The significance of such results reflects a possibility that some BVs, at specific locations, are shared by more than a single scale. However, the response of $WB\left(x,y\right)$ should yield a binary image and, therefore, should yield only a response of zero or one. In order to overcome this issue, we chose to use a criterion that yields a response that acknowledges what will be the value at a specific location under the below conditions; see Equation (29). The criterion can be taken as the summed value τ (Methods), which reflects a chosen $\hat{WB}\left(x,y\right)$ that will determine if $\hat{WB}\left(x,y\right)$ will be finally determined as one or zero.
(29)$$\hat{WB}\left(x,y\right)=\left\{\begin{array}{c} 1\;if\;WB\left(x,y\right)>\tau \\ 0\;otherwise \end{array}\right.$$
where τ is the intensity threshold that will end up with final values for $\hat{WB}\left(x,y\right),$ which will determine the ROI image $\hat{WB}\left(x,y\right)$.

### 3.4. BV Texture

The output of the enhanced edges, $T_{wb},$ (the line completion responses) in the image $T_{E}$ (Figure 3C, Equation (23)) can be referred, in this stage of the algorithm, only to the labeled ROI (Figure 4C2, Equation (30)). Therefore, $\hat{WB}\left(x,y\right)$ (Equation (30) and Figure 5) is calculated to delineate the spatial location in which the line completion $T_{E}$ is applied.
(30)$$T_{wb}=\hat{WB}\left(x,y\right)\times T_{E}$$

### 3.5. BV Visibility Enhancement

The BV edge image, $T_{wb}$, contains the enhanced edges within the relevant ROI’s locations, but this image is not contained in the diverse DC details, which are not at the edges, such as in the input image $I\left(x,y\right)$. Before performing the integration of the edge image $T_{wb}$ with the original image, we need to deal with the separation of the positive $T_{P}$ and negative $T_{N}$ values of the edge image $T_{wb}$ (Equation (32)). The integration operation is completed also to enable an appropriate correction of the dynamic range of the final image $I_{E}\left(x,y\right)$ (Equation (33); [27]).

The separation of the blood vessels’ edge image $T_{\mathrm{wb}}$ into positive $T_{P}$ and negative $T_{N}$ responses is shown below:(31)$$T_{P}=\max\left(0,T_{\mathrm{wb}}\right),\;T_{N}=\max\left(0,-T_{\mathrm{wb}}\right)$$where${\;T}_{P}\;\mathrm{and}\;T_{N}$ represents the absolute values of the positive and negative responses through the maximum operation.

The following stage, consequently, is the integration of the texture image of both the positive, $T_{P},\;$and negative$,\;T_{N},$ blood vessel responses (Equation (32); Figure 6), with the input image $I\left(x,y\right)$ (Equation (33)), Figure 6. The weight function of each of the components, the original image, and the edges, can be determined by factor α. In order to improve the dynamic range image, we need to have a similar range of values for the positive and the negative components of the image, and this is achieved through the β factor, as shown in Equation (33). This β factor is intended to enable the balance of the bright and dark zones, for the $I_{E}\left(x,y\right)$ response.
(32)$$I_{E}\left(x,y\right)=I\left(x,y\right)+\alpha \times \left[\frac{\;T_{P}}{1+\beta \cdot I\left(x,y\right)}-\frac{T_{N}}{1+\beta -\beta \cdot I\left(x,y\right)}\right]$$
where each polarity component is balanced by factor β. α,β are positive constants. The aim of the division factors is to suppress the positive or negative responses that are near the zero clipping, respectively.

## 4. Methods

### 4.1. Dataset

Our clinical dataset consists of coronary cine-angiograms acquired during 3 different routine interventional catheterization procedures performed on 3 different patients in a single medical center. Altogether, 62 different runs were acquired, with an average of 40 frames per run. The overall number of coronary angiogram frames in the dataset is 2500, and the resolution of each image is 1024 × 1024 pixels, with 1024 grey levels per pixel. Our implementation and testing were performed in a MATLAB environment, using an $Intel^{\text{®}}\;Core^{\text{™}}$ i9 2.3 GHz CPU laptop with 16 GB of main memory. The average processing time is 1.9 ±0.05 s per frame.

### 4.2. Model Parameters

#### 4.2.1. Orientation and Scale Filtering (Gabor)

The Gabor filter (Equation (1)) parameters are σ=8, λ=12.

Instead of using multiple Gabor filter sizes, the algorithm uses a fixed Gabor filter size, while deconstructing the input image into multiple scales, as is commonly used in Gaussian Image Pyramid deconstruction. The chosen number of scales is 8, and the number of Gabor filter orientations is 8. The chosen orientation angles, θ, are  as follows:0°–360° (Equation (1)).

Downscaling and upscaling of the images were completed by using the Lanczos resampling method. This method approximates the theoretically optimal reconstruction filtering for band-limited signals, which uses the Sinc filter. We used it for its anti-aliasing and preservation of sharp edge properties, which are important for maintaining the quality of the images, along the different algorithm stages.

The parameters used for the Naka–Rushton function (Equation (8)) is $\sigma_{NR}^{n}=0.1,\;n=2$.

The parameter used for the orientation recombination of the different images is m1=1 (Equations (20) and (21)). The parameters used for the scale recombination of different images are m2=2 and γ=1/8 (Equations (22) and (23)).

#### 4.2.2. Line Completion Algorithm

The values of $\sigma_{x},\sigma_{y}$ (Equations (10)–(12)) represent the magnitude of the additive signal. Equation (10)’s decay in the collinear axis and its orthogonal weight function of the signal, Equation (11), are defined separately for each scale (Table 1). The percentile used for the threshold of the line completion, $\gamma_{j}$ (Equation (18)), is also detailed in Table 1.

#### 4.2.3. BV ROI

The structuring element disk radius, $r_{j}\;\left(pixels\right)$, is used for the closing procedure (Equations (25) and (26)) as detailed in Table 1. The width of the median filter is L=3; see Equations (27) and (28). The resulting number of median-filtered scales, $\hat{wb_{k}}\left(x,y\right),$ is 6.

#### 4.2.4. BV Texture

The parameters used for integrating the enhanced structure signal and the original image are α=0.4 and β=4 (Equation (32)).

### 4.3. Cardiologists’ Evaluation of the Algorithm Results

All images were assessed by the consensus of 2 experienced interventional cardiologists (authors of our paper). First, viewing Diacom format cine runs of clinical coronary angiograms and choosing from each run (15 frames/s), we acquired, from standard routine angles, a single frame that best demonstrates the most significant clinical procedural details. These included general visibility of the whole frame, the CBVs, especially the small ones, the lesions within the BVs, and the various angiography interventional tools. Next, each cine run and each single chosen frame (“raw” image), as above, were visually compared, first with the same image after further manual correction for optimal visibility (modifying brightness and contrast) and, finally, with the improved results and the post-processing results of the novel algorithm.

## 5. Results

The algorithm results are presented according to the clinical implications. The figures in the current paper represent a partial sample from the overall results, which were selected by the cardiologists, mainly due to their diagnostic significance.

### 5.1. Overall Visibility Enhancement

Figure 7 demonstrates the overall improvement in the visibility of the BVs in the images derived from two different CAGs, from two patients. The result images show that the overall visibility of the BVs is enhanced with enhanced appearance. It can be seen at a naïve glance that the algorithm improved the dynamic range of the images, and therefore, the illumination and brightness of the images are more suitable for observing the structure and the richness of the vascular tree. The cardiologists who assessed the algorithm results considered the general visibility of all the CBVs as improved, with special emphasis on the better visibility of the small BVs, the collaterals, and the overlapping side branches (Figure 8).

### 5.2. Blood Vessel Interiors

One of the goals of our algorithm was to enable observing the BV’s interior content and expose important clinical details, such as the catheter tip, catheterization apparatus, and inflections. Figure 8 (right column) demonstrates the exposure of the interior part of the major CBV, as represented by the long arrows. The cardiologists assessed the better visibility of the content within the lumen (the interior of the vessel) of the large BV, which enabled somewhat better identification of overlapping vessels (versus similar-looking luminal lesions) (Figure 8b, long arrows), coronary artery dissections (Figure 8c, long arrows), various interventional tools such as the tip of a diagnostic catheter (Figure 8a, long arrow), wires (Figure 8c, short arrow; Figure 9c, left arrow), balloons (Figure 10b,c long arrow), and stents (Figure 9a, long arrows; Figure 10a, long arrows).

### 5.3. Small Blood Vessels

A successful diagnosis by cardiologists requires proper observation of most of the blood vessels, including the small ones (Introduction). Figure 9 and additional figures, such as Figure 7 and Figure 10a, demonstrate the exposure of a single or a net of small BVs with deficient exposure in the original image. Figure 9 demonstrates the ability of the algorithm to enhance the visibility of small BVs, which is of special clinical importance. The better exposure of a net of small BVs that feeds the interventricular septum (Figure 9a2, right arrows) can indicate a successful intervention in the right coronary artery stenting and replenishment of blood supply (Figure 9b2, left arrows). An enhanced small BV can be seen in Figure 9b2, where a collateral from the left to the right coronary artery is demonstrated. This helps to diagnose, from left system imaging, a proximal obstruction of the right CBV, since this collateral is not usually seen where the right CBV has good blood supply.

### 5.4. Low-Contrast Agent, Low-Radiation, and No Contrast Agent

We also tested angiography images with low radiation and low contrast, with the same algorithm that has the same set of parameters, which was used for the whole set of CAG images.

Our dataset lacks true low-contrast angiography procedures. Yet, we identified low-dose conditions where the injected contrast agent barely arrives at its specific locations or where the contrast agent starts to wash out and is partially cleared from the BVs. Thus, for this purpose, we used timed CAG images where the contrast agent is diluted in the BVs.

Figure 10 represents several examples from our dataset that can be regarded as acquired at low radiation (Figure 10c), at low contrast (due to dye clearance, Figure 10a), and in the absence of a contrast agent. The last group of images was acquired before the contrast agent injection (Figure 10b). Figure 10a1 presents an example of an image acquired during the clearance of the dye from the proximal part of a large CBV. VIAEVCAs enabled improved visibility of the major CBV content and the stents (Figure 10a2, long arrows). The algorithm also succeeded in enhancing the visibility of the small CBVs, although the dye started to be cleared distally at this stage.

Figure 10b refers to the CAG condition where the contrast agent was not yet injected into the BVs, and therefore, only the catheterization tools are exposed. The algorithm better exposed the wire in the internal side of the diagnostic catheter (left arrow) and inside the CBV and the attached balloon (perpendicular arrow). In this image, the artifacts of the bony structure are also exposed (thick and short arrows.)

The reduction in radiation to 1/13 per image (attempting to reduce tissue damage for the documentation of stent deployment) did not impair the advantage of the algorithm to enhance visibility (Figure 10c2), compared to the natural camera image (Figure 10c1) or even manually processing the same image in an optimal manner. Note also that the algorithm partially performs a compounding compression and expands the dynamic range of the image. This is expressed by the exposure of the image information in the too-dark and too-bright regions (this can be seen across all images).

Notably, although the algorithm succeeded in improving many aspects of the CAG images, it also enhances the appearance of disturbing vertebral and bony artifacts, which can appear as small blood vessels and may lower diagnostic accuracy, especially while viewing only still images (Figure 8a2 and Figure 9a2). However, viewing the images from other camera angles and in the cine mode, which allows the evaluation of flow within the CBVs, mostly helped to clinically differentiate between those artifacts and true small vessels (Figure 10a2).

## 6. Discussion

Our novel algorithm has been inspired by the visual system mechanism for contour integration and lateral facilitation [27,29]. It succeeds in overcoming several critical CAG appearance and visibility challenges, such as small BVs, catheterization tools, and visibility under low-radiation and/or low-contrast agent conditions. The enhanced CAG visibility (Figure 7, Figure 8, Figure 9 and Figure 10) led to improved images for better identification and classification of obstructive lesions, collateral blood vessels, the evaluation of flow, tissue perfusion, and catheterization tools. The algorithm also revealed an exposure of unwanted details of bony structures, which may interfere with the diagnosis of the small BVs. This drawback is expressed mainly while viewing still images acquired from limited angles but much less in the cine mode and at multiple angles.

The enhanced CAG visibility suggests a better decision-making process by the expert cardiologist before, during, and after the procedure. These achievements have been obtained with a single compound algorithm, which also succeeds in significantly enhancing images even with a low-dose contrast agent or low radiation, while simultaneously reducing noise and illumination problems. The advantages of the algorithm’s performance are compared to the general and topic results of previous studies.

Small blood vessels: Several recent studies related directly to the issue of revealing small BVs [2,8,14]. Some previous studies related to the issue of small BVs as only a side effect of generally enhancing the image, through different methods, such as de-noising [9,15,19]. The noise reduction succeeded in enhancing the appearance of BVs, but it appears that the exposure of small BVs was not salient [9,15,19].

Two of the more recent segmentation studies applied a deep learning U-net [14] and decision tree [8] and succeeded in obtaining better results in reference to small BVs. This was achieved by cardiologists manually delineating this type of BV for the training dataset. Still, the visual impression of the small BV segmentation was considered insufficient [8,14].

It appears that the VIAEVCA exposure succeeded in revealing a larger repertoire of small BVs (Figure 7, Figure 9a2, and Figure 10a), along with bony artifacts, which, however, can be partially overcome, as described above. These artifacts are indeed prevented when a routine segmentation operation is performed. Nevertheless, our algorithm’s enhanced visibility provides a much better ability to observe the small BVs, an important diagnostic advantage. The common goal of a successful invasive procedure is usually the restoration of flow in the obstructed CBV, especially under emergency circumstances such as acute myocardial infarction (AMI) or acute coronary syndrome (ACS). Yet, sometimes, despite epicardial patency after opening the artery, the restoration of good flow and myocardial perfusion is absent. This no-flow phenomenon has significant clinical importance for treatment and prognosis, but the current methods to clinically assess it (measuring or scoring flow and “myocardial blush”) are limited, time-consuming, or expensive.

Future research may show a correlation between this no-flow phenomenon and the disappearance of small vessels, which were visible before the invasive intervention only by using the algorithm. Also, cardiac diseases such as syndrome X or metabolic cardiac syndrome, which are defined by typical anginal pain with myocardial ischemia but normal patent coronary arteries, may eventually show abnormal small vessel disease, detectible due to the enhanced visibility offered by the current novel algorithm. We could not find other algorithms that relate to these pathological issues.

Since previous algorithms [24,25] produced segmentation of the BVs, they could provide a quantitative measure for accuracy performance (in comparison to the manual cardiologist segmentation). One study [24] arrived at a 95% accuracy performance, based on seven images, which were examined. Ref. [14] used a larger dataset consisting of 48 images and arrived at a DICE performance of 76%. Our structure enhancement algorithm results cannot be compared quantitatively to the segmentation results since our algorithm does not produce BV segmentation. In fact, even the high DICE or accuracy performance, as above, does not necessarily reflect the results related to small BVs, since the area that they occupy in the image is relatively small. Additionally, the above studies [24,25] based their results on single images instead of cine images, which are commonly used in diagnostic CAGs. Our algorithm enabled clinical assessments based on consecutive images and the cine mode of the images, which can also reduce the effect of distractions caused by the bony artifacts.

Several groups presented algorithm results using the CNR measure. Lee and his colleagues [17] showed an 81% CNR improvement over the low-dose images (acquired using 2% of the radiation used in normal CAGs). They arrived at a higher CNR measure than our results, a 60.71% CNR improvement. They achieved a higher CNR score probably due to the noise reduction that better smoothed the background image but did not necessarily expose a larger number of small BVs (their images do not show a trend of a higher number of small BVs).

Studies that performed segmentation by background subtraction [18,20,21,22] yielded much better CNR results (~200% CNR improvement over the original) than studies that performed enhancement of visibility, as ours. This was achieved by the isolation of the BV image from the background and the obtained separation from other organs, such as the spinal cord, and probably, due to the fact that different organ components have different spatial-temporal subtraction properties [18,20,21,22]. The price for such sophisticated algorithms was the algorithms’ complexity and performance durations. In any case, it is expected that the segmentation operation will lead to a criterion with a much higher CNR measure, which may not necessarily reflect all clinical demands and cannot really be compared to our visibility measure.

We further ask whether the measure of CNR, which reflects different average intensity values in a specific region in the BV in comparison to the background, is, indeed, an efficient measure for better visibility of the BV. It could be a type of legitimate measure if the only need would be to enhance the visibility of the major BV, but by increasing the visibility, we would like to achieve an additional two measure goals. One of them is to enhance the visibility of small BVs, for increasing diagnostic abilities (Section 4.3). This goal was reported by just one of the background reduction studies (Thuy et al., 2021 [25]). The second goal is to increase the visibility of the BV content, such as the CAG tools and plaques, enabling improved procedures. This BV exposure was probably derived from increasing the visibility of the texture inside the BV region (i.e., increasing the variance values) and not simply increasing the intensity values, as utilized by segmentation operations. Therefore, a common CNR improvement cannot be reflected by such a required property of the background subtraction results and their CNR-obtained measures. Nevertheless, future studies may combine these methods to enhance BVs in addition to improving visibility inside vessels using various CAG tools.

It has been reported by one background reduction study that, despite these algorithms’ impressive results, the exposure of small BVs is still lacking [20], and also, the visibility of the catheterization tools inside the BVs is missing. Furthermore, these methods can lead to a condition where the small BVs can be fragmented into different layers (background and foreground, for example), due to matrix decomposition [20]. These authors suggested the application of Gabor wavelets to enable the enhancement of the tiny blood vessels [30] but did not test this possibility. Our algorithm can be used as a type of preprocessing for image enhancement and might be more beneficial than a Gabor wavelet (ref), due to its ability to perform line completion while decreasing the noise level.

Low Dose: A comparison of our algorithm results on low-dose conditions (low-radiation and low-contrast agents) is not sufficiently accessible, since there is only one paper [17] that provided algorithm results regarding low radiation at the CAG images. An additional two papers, which improved the visibility of the BVs in conditions mimicking low-dose radiation [1] and a low-contrast agent [18], showed an improved image with somewhat better visibility of the BVs.

To evaluate our algorithm results on the low-radiation images, we used images taken from a fluoroscopy run (used for the documentation of stent deployment), which routinely utilize only 1/13 of the radiation dose, relative to CAG images (Figure 10b2). The expert cardiologists assessed the algorithm results, which succeeded in enhancing the visibility of the procedural tools, such as the catheter tip, the wire, the balloon, and the stent edge markings (Figure 8, Figure 9 and Figure 10). None of the previous studies showed such an ability to obtain visibility of the inner BV tools, neither under normal radiation and contrast agent doses nor at low-radiation or low-contrast agent conditions. We, therefore, propose our algorithm, which shows a potential usage of a CAG based on a low-radiation or low-contrast medium, which is less medically harmful. Future clinical studies using the algorithm are expected to validate the method and enable longer procedures, as needed, especially for more patients susceptible to damage, such as patients with renal dysfunction.

Inner blood vessel visibility: In the literature, there is a reference to catheter visibility and detection, but only outside the BVs [11], with no reference related to enhanced visibility of the catheterization tools in general. More specifically, previous algorithms that performed different algorithms for contrast enhancement, such as by fractional differential kernels [15] or Hessian matrices [19], did not refer to those tools.

We also tested whether an algorithm that has the potential to expose the dynamic range of specific areas in the image, such as the CLAHE algorithm (Matlab), has the potential to expose the catheter tools in the CBVs. We found some BV enhancement, but with significantly reduced visibility of the tools than our algorithm presents.

The VIAEVCA algorithm is probably the first to enable distinct enhanced visibility of the CAG procedural tools inside the major BVs. This achievement was obtained through an innovative approach to object enhancement utilizing the lateral facilitation mechanism component and additional computations regarding BV texture. Since our study, in its present form, does not perform segmentation or subtraction operations, it cannot compete with published background subtraction results and their CNR-obtained measures. Nevertheless, future studies may combine these methods to enhance BVs in addition to improving visibility inside the vessels using various CAG tools.

## 7. Conclusions

The present study presents a novel algorithm for enhancing BV visibility from routine angiography video images.

Original algorithm: (a) This algorithm was inspired by an educated guess mechanism suggested for the visual phenomenon of lateral facilitation. It causes line and texture completion imitating our visual perception and is expressed by the lateral facilitation phenomenon. (b) The novelty of the algorithm is also expressed by incorporating partial noise reduction simultaneously with line completion, as part of strengthening the structure.

Algorithm achievements: (a) This algorithm enhanced the visibility of the BVs, including the small BVs. (b) This is the first algorithm that demonstrates enhanced visibility of major BV luminal content, including CAG procedural tools and plaque lesions. (c) Through the algorithm adaptivity, it also succeeds in enhancing BV signals at low doses of contrast agents and radiation, with the same set of parameters. These achievements contribute together to support the clinician in making an accurate diagnosis, better procedural performance, and improved patient outcomes.

## Figures and Tables

**Figure 1 biomimetics-10-00018-f001:**
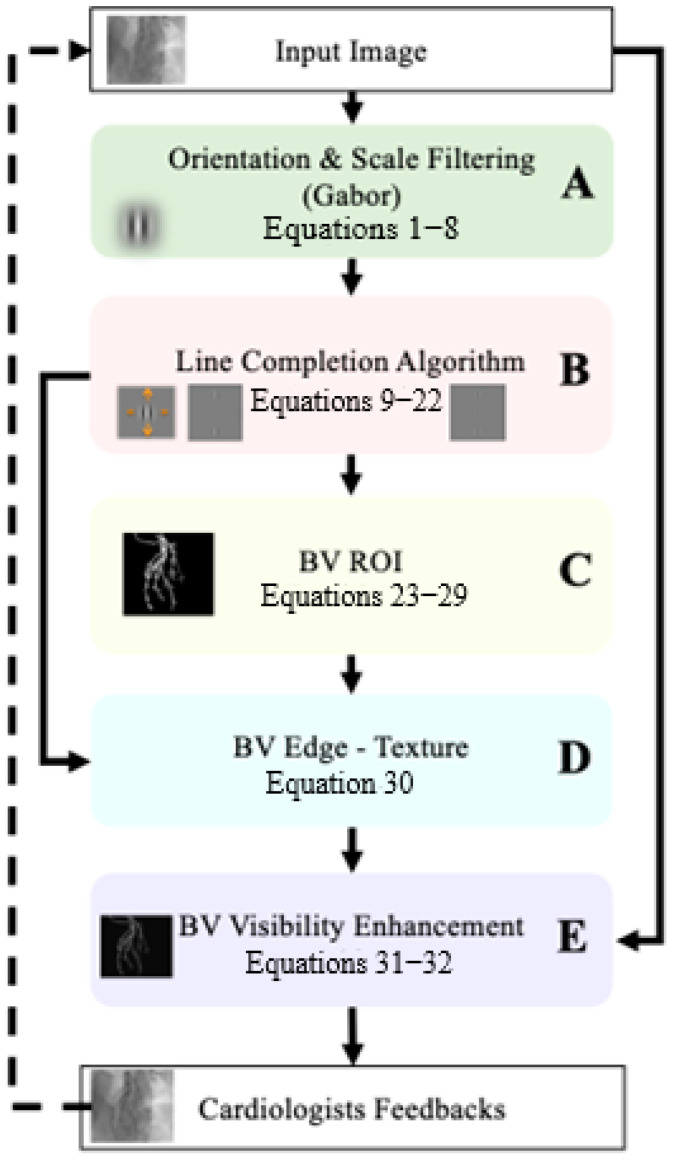
The schematic building block of the main components of the VIAEVCA algorithm. Each colored block represents a different algorithm component (with reference to the appropriate equation numbers in the model). The same code of block colors is given for the different algorithm components and will be used for the rest of the schematic building blocks presented along the algorithm components description below.

**Figure 2 biomimetics-10-00018-f002:**
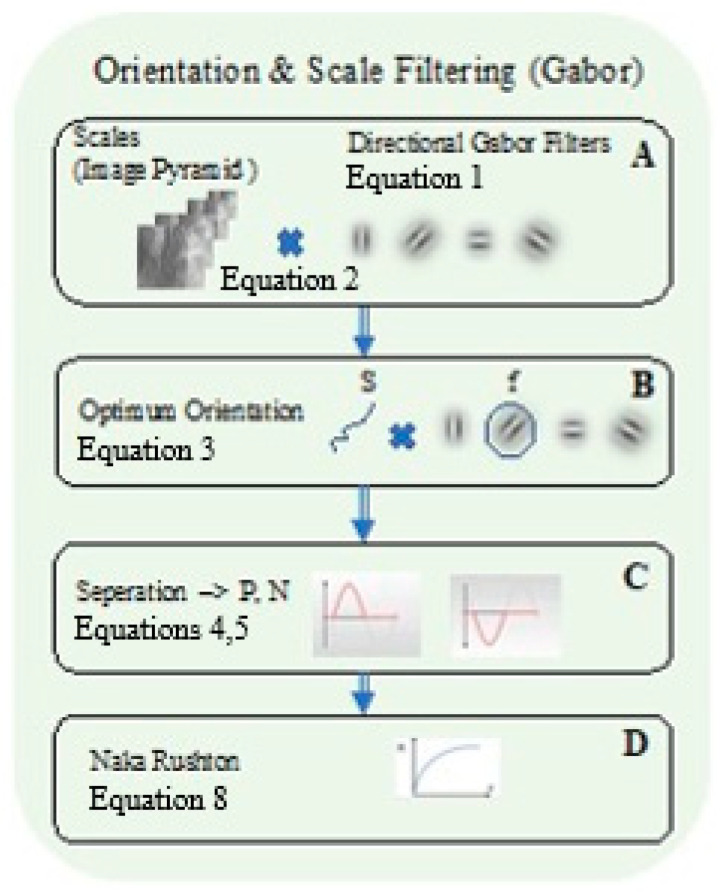
Schematic block diagram of the initial part of the algorithm that computes the different Gabor receptive fields of orientations and spatial resolutions. (**A**) Direclioral Gabor Filters. (**B**) The optimal orientation and scale filters are then computed. (**C**) The negative and positive components are separated. The small plots on the right and left sides illustrate the mathematical operations. (**D**) The Naka–Rushton operation.

**Figure 3 biomimetics-10-00018-f003:**
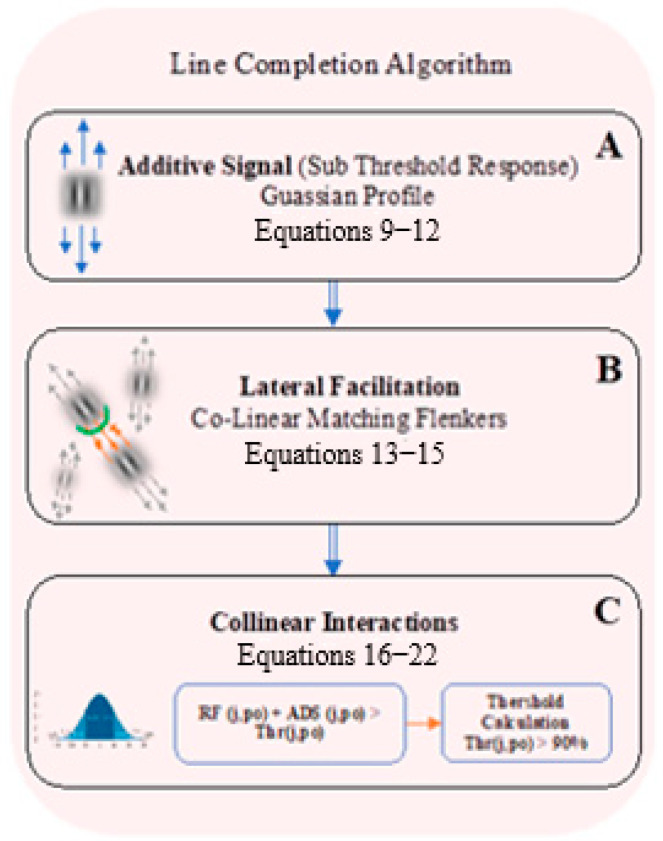
Schematic diagram of the line completion algorithm’s component. (**A**) represents the additive signal, which refers to the extra-classical receptive field area that is located collinearly to the receptive field (the blue arrows). (**B**) illustrates the case that lateral facilitation is fulfilled when the additive signals of collinear responses (grey arrows) are overlapped (orange arrows). The green half-circle represents the weight function for the directional facilitation between adjacent receptive fields, through a Gaussian profile; see text. (**C**) illustrates the threshold response that was determined by the percentile response (blue Gaussian, **left**).

**Figure 4 biomimetics-10-00018-f004:**
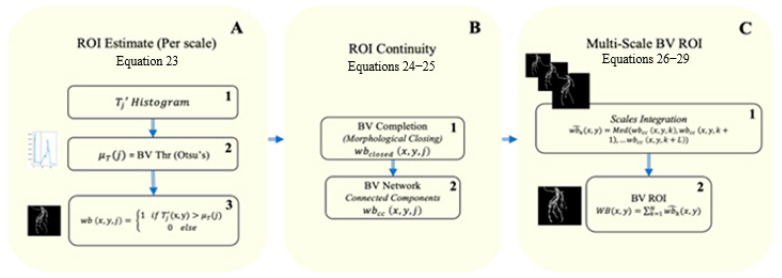
Schematic diagram of the blood vessel ROI detection, through three main stages. (**A**) illustrates the acquired binary ROI estimate (3), through the application of Otsu’s threshold method on the *LF*(*j*) “image” (1–2) (left blue histogram). (**B**) represents the calculations of closing and connected operations. (**C**) represents the integration of the different response scales to compose the overall blood vessel’s ROI (2).

**Figure 5 biomimetics-10-00018-f005:**
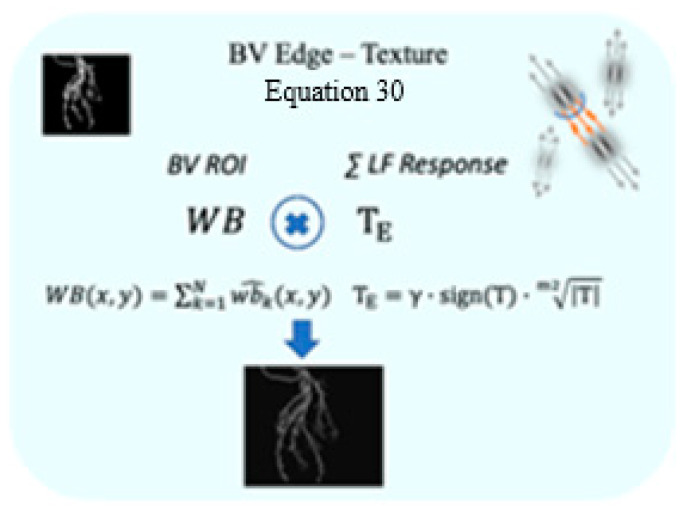
The flow chart illustrates the multiplication of the extracted ROI (**left** binary image) with the multiscale *LF* response. The right illustration demonstrates receptive fields with their summated additive signal (orange arrows). The figures represent the extracted ROI (**left** upper image) and the resulting blood vessel multiscale. edges (**bottom** image).

**Figure 6 biomimetics-10-00018-f006:**
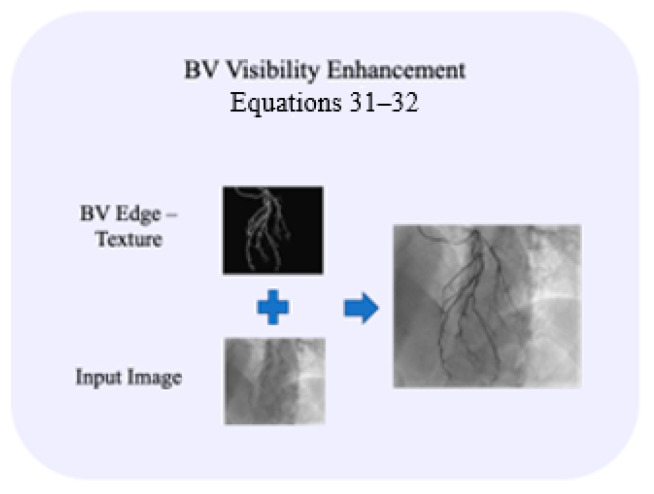
Schematic diagram of the last stage of the algorithm that integrates the edge image (**upper left** image) and the input image (**bottom left** image) into an enhanced visibility image (**right** image). An additional stage is performed, which is related to the dynamic range.

**Figure 7 biomimetics-10-00018-f007:**
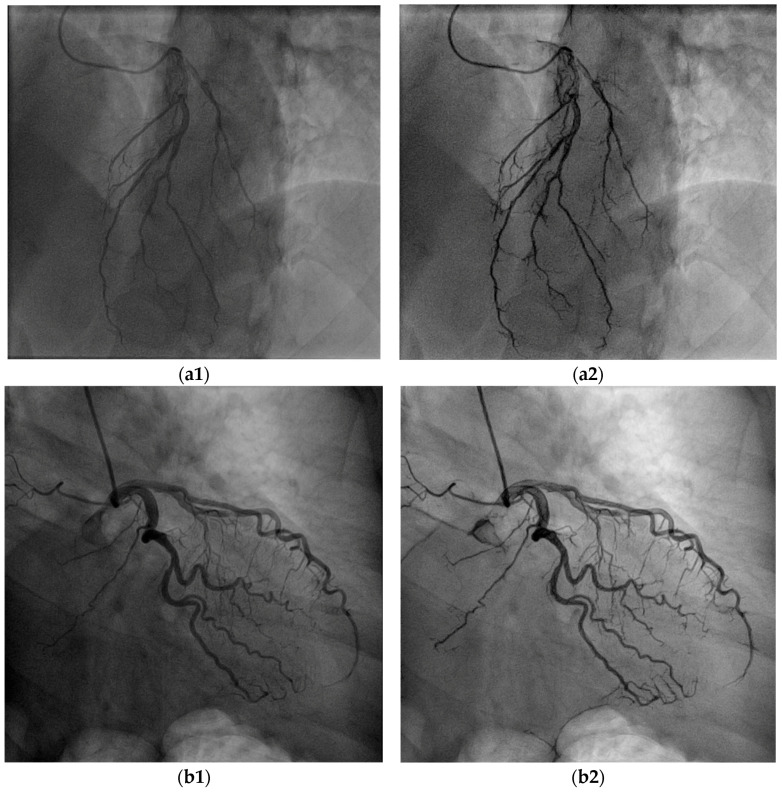
Two examples of VIAEVCA results (**a2**,**b2**), which represent the better overall appearance of an angiogram with a better view of the BVs, including of the small ones. Original (“raw”) images are shown on the left column and the algorithm results on the right column. This mode of presentation of the image results refers to all the figures in this study. (**a1**,**b1**) are the original images before the algorithm’s applications.

**Figure 8 biomimetics-10-00018-f008:**
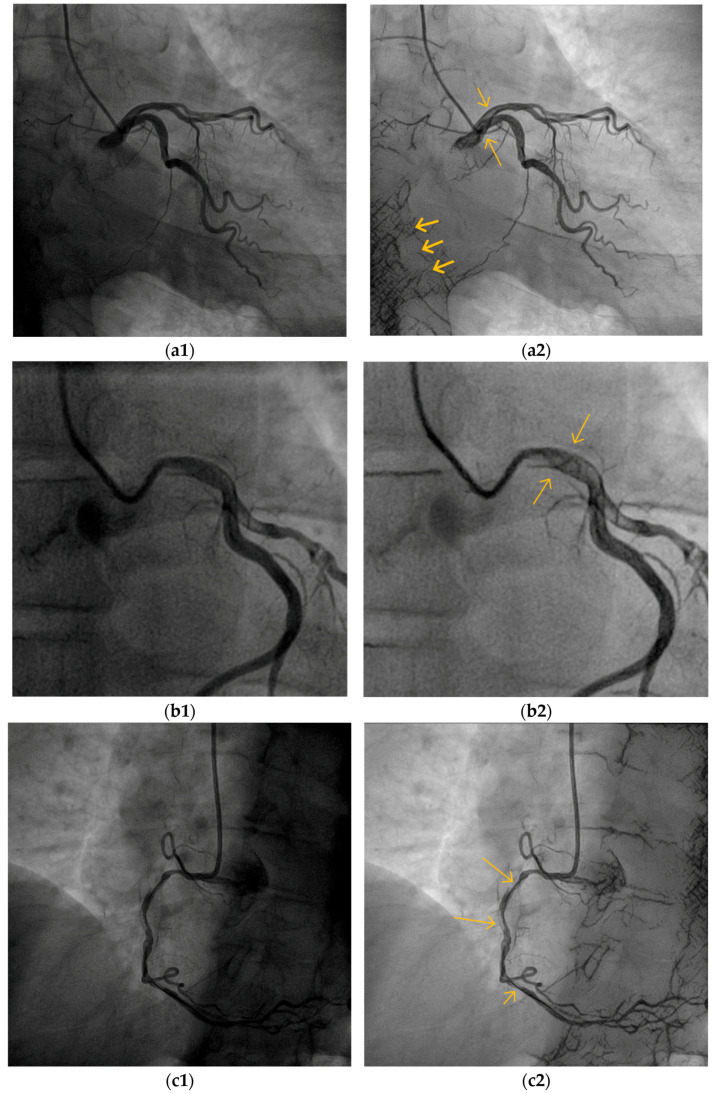
The VIAEVCA succeeds in increasing the visibility of the interior content of the BVs while revealing important clinical details. (**a2**) The upper long arrow indicates the exposure of the catheter tip, while the upper short arrow indicates exposure of the inner lumen of the left main coronary artery. The spine artifacts are exposed and are indicated by the thick left arrows. (**b2**) The BV shows overlapping vessels and lesions (see text); see the long arrows. (**c2**) Two dissections of the right coronary artery (long arrows) are exposed. The wire is also more visible (short arrow) with the results image. (**a1**,**b1**,**c1**) are the original images before the algorithm’s applications.

**Figure 9 biomimetics-10-00018-f009:**
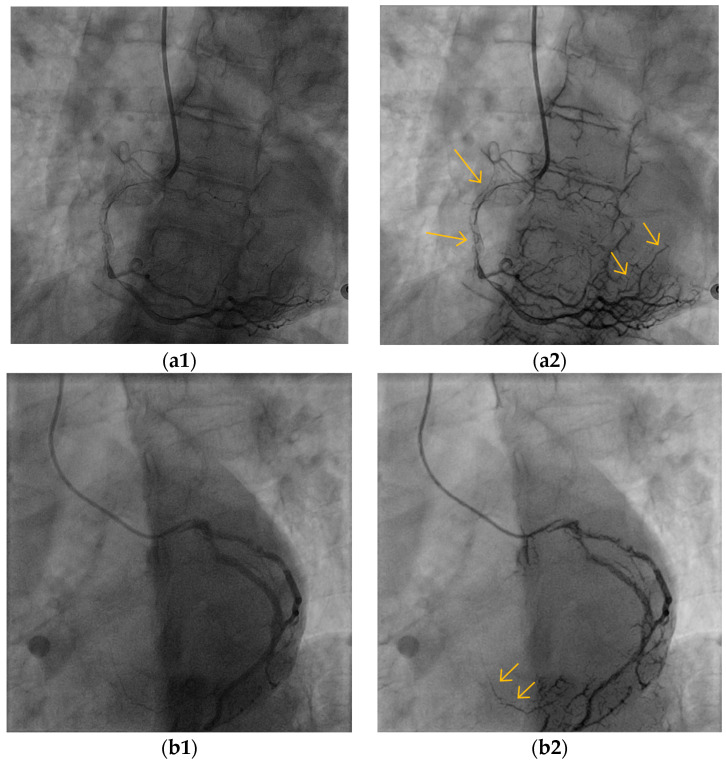
The VIAEVCA enables the better appearance of the small and clinically important blood vessels. (**a2**) The right arrows show better visibility of the small BVs, which split and climb upwards. Their diagnostic significance is described in the text. The left arrows show the two stents. (**b2**) The arrows show the exposed right coronary artery (see text for clinical significance). (**a1**,**b1**) are the original images before the algorithm’s applications.

**Figure 10 biomimetics-10-00018-f010:**
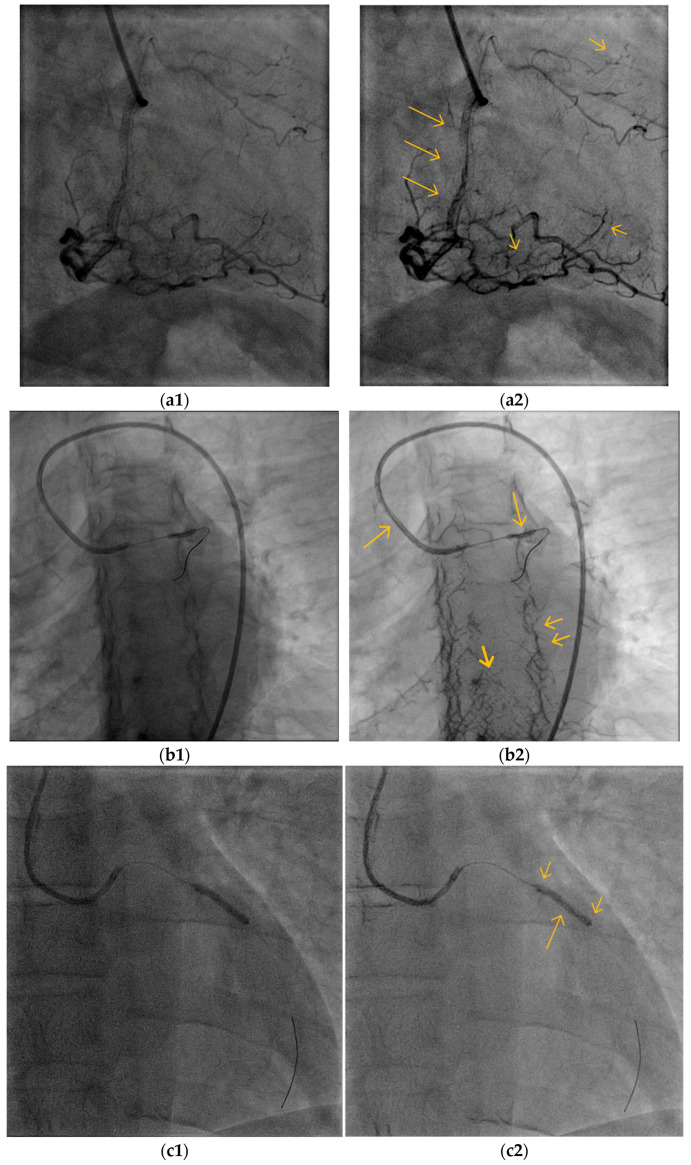
The VIAEVCA succeeds in exposing clinically significant knowledge, even at low-contrast agent (**a**,**b**) and low-radiation scenarios (**c**). (**a1**) shows a low-contrast agent image (due to dye clearance). (**a2**) The VIAEVCA succeeds in enhancing the visibility of both the stents (left long arrows) and the small blood vessels (right short arrows). (**b1**) shows an image before contrast agent injection. (**b2**) The VIAEVCA succeeds in showing the wire inside the diagnostic catheter (left long arrow) and inside the CBV with the inflated balloon (right long arrow). Bony artifacts (thick and short arrows) are exposed in the region of the spine along with accentuation of edges of spinal bone structures. (**c1**) shows a low-radiation image (1/13 reduced radiation). (**c2**) The stent with its proximal and distal metal markings (short arrows); the wire that passes through it (long arrow) is better exposed.

**Table 1 biomimetics-10-00018-t001:** Algorithm parameters per scale.

Resolution Index (*j*)	$\;{\sigma_{x}}_{j}$	$\;\sigma_{y_{j}}$	$\;\gamma_{j}\;\left[\%\right]$	$\;r_{j}\;\left(pixels\right)$
1	0.05	0.05	99	0
2	0.05	0.05	80	2
3	0.06	0.06	97	3
4	0.07	0.07	97	4
5	0.07	0.07	96	5
6	0.07	0.07	96	5
7	0.1	0.1	96	5
8	0.1	0.1	96	6

Part of the algorithm’s parameters has a different set of values per scale (*j*). These parameters are presented in Table 1. The background color has been added for identification the used resolutions for small vessels inside the cardinal vessels.

## Data Availability

The original contributions presented in this study are included in the article. Further inquiries can be directed to the corresponding author.

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
