# Peer review of "Visual System Inspired Algorithm for Enhanced Visibility in Coronary Angiograms (VIAEVCA)"

_biomimetics, 2025, doi:10.3390/biomimetics10010018_

Round 1

Reviewer 1 Report

Comments and Suggestions for Authors

The article introduces a biologically inspired algorithm that enhances blood vessel visibility in coronary angiograms, especially in challenging conditions like low radiation and contrast. The suggested solution effectively combines lateral facilitation principles from the visual system, using Gabor filtering and line completion models to make it easier to see the vessel and lower the noise. The paper is very interesting.
I still have some comments, questions, and suggestions:
C1 : The evaluation is based on a relatively small dataset (62 runs across 3 patients), which may limit the generalizability of findings.
C2 : The implementation of the algorithm may necessitate substantial computational resources and particular expertise, potentially limiting its practical adoption.
Q1 : How does the algorithm manage overlapping vessels in areas of high vessel density?

Q2 : Is it possible to dynamically optimize the algorithm's parameters for varying patient datasets or imaging devices?

Q3 : What is the potential influence of artifacts on diagnostic decisions, and how can this be reduced?
Q4 : Are there intentions to incorporate this algorithm with AI-driven tools for automated vessel labeling and diagnosis?
S1 : Section 6 is very interesting but very difficult to follow. While keeping its content, it is imperative to summarize the entire section in a clear table including all the aspects of comparisons raised.

Author Response

Agreed,. Thank you, we amended the manuscript according to your comments. Attached here are our responses

Reviewer 2 Report

Comments and Suggestions for Authors

Summary:
This study proposes an algorithm pipeline the authors call VIAEVCA with the goal of improving heart blood vessel imaging. The technical problem is treated as one of line enhancement while attempting to minimize unwanted artifacts. The authors summarize the last 10 years of literature on CAG image improvement algorithms while noting that this work is not intended to be a comprehensive literature review. The algorithm uses a classic multiresolution approach to mitigate sensitivity to image feature size and kernal size across the various mathematical operations used. The initial group of algorithms are run in many orientations, too, so as to capture features oriented in most 2D directions. There is an intermediate segmentation and masking step meant to select around the blood vessels to be enhanced while keeping the background unchanged. One of the innovations of the paper relates to the line completion. The authors combine an additive signal (a fairly classic approach to the problem) with co-linear matching flankers. My way of thinking about this is a hypothesis field where there is a greater 'chance' of a match in some situations versus others. The authors use the terms lateral field and receptive field instead. I think this is intended to help with the challenge of line completion in the realistic and difficult scenario of low contrast coronary angiography. The authors test their method qualitatively by asking two interventional cardiologists (who are authors on the paper) to point out areas where the images are better or worse than the original images. They conclude that their algorithm pipeline is effective despite some intrinsic challenges when other features are present in the images like bone or image underexposure.

Opinion:
I think this work is overall successful at achieving its goal. The images presented in the paper and presentation of the pipeline convince me that the method works as the authors hoped. The pipeline of algorithms is pretty long and no source code is provided. This would make a new implementation pretty challenging I think for most people. I see this as a major limitation of the proposed impact of the work. Another critique is that the qualitative assessment is going to be biased because the image reviewers are authors and it was not stated that the study was blinded. So I think it is presented in a somewhat misleading manner, as the authors use language similar to that of studies where the reviewers or subjects are blinded to the origin and nature of the images.

Strengths:
-there is a huge amount of literature in this area of research. To avoid excessive citations, the authors state that they will focus on recent advancements and studies. Being explicit about this is a good thing.
-The summarizing figures for algorithm steps are helpful for reader comprehension.
-The images provided in the paper illustrate improved dynamic range and distinction of blood vessels from the background
-The grammar is generally good with only some small errors noticed. A few of these are listed line by line at the end of my review.

Weaknesses:
-the source code for the paper is not provided making the work unnecessarily difficult to reproduce
-the reviewers of the image data were also co-authors and this is not disclosed until very deep into the paper
-the reviewers of the image results were not blinded to the conditions nor context of the image data and thus there will be a conflict of interest and bias
-The algorithm pipeline requires many parameters to be chosen (i.e., sections 4.2.1 and 4.2.4). It was not obvious what might be the impact of poorly chosen parameters on the resulting output image. While I am not requesting a full parameter sensitivity study, perhaps the authors could comment on what led to the choices that were made and what might be the outcome if poor choices are made.

Areas for Improvement:
-If the authors want to make statements about the quality of the images, then just do that without saying that the work was reviewed by two experts. This implies that the experts are not related to the study and don't know the context of the work. But that is not the case here. So I would revise the text to make it more clear that these are your opinions about the quality of results and not those of independent reviewers. If independent review is the goal, this should be handled with more care - more people should be involved, they should not be related to the study, and they should be blinded to the context and treatment of data.
-The source code could be provided in a public repository (Zenodo, github, Dryad, etc.). I think there should be no disadvantage to doing this, so I see it as a clear area for improvement.
-It may be interesting to test the impact of the algorithm pipeline on other data types. This will make it more clear to an image processing expert (I think this is the intended audience considering the level of math presented) what is the effect of these filters and the parameters that were chosen.

Line Item changes:
35 - remove period after "2020)"
56 - replace "Such is" with "Such as"
67 - remove extra ")"
72 - change "have been also vastly used for both" to "have commonly been applied for"
158 - the abbreviation "RFs" is used here without being defined. I think it means Receptive Field, as it is defined much later on 262-263. Either define it here or don't use the abbreviation. I think this abbreviation is also used as a subscript in 240, 247, 248, and Eq 8 before being defined.
Fig 3 caption - last line has open parenthesis.
Fig 3 - the word "Flenkers" is used, but I think the authors meant "Flankers"
264 - a letter is underlined for no obvious reason
Fig 7 caption - replace "reffers" with "refers"
623 - replace "patent" with "patient"
820 - replace "via EVCA" with "VIAEVCA" to be consistent

References to consider:
n/a

Author Response

Agreed, thanks. Attached here are our responses according to your generous comments.

Round 2

Reviewer 2 Report

Comments and Suggestions for Authors

The authors incorporated or responded to most feedback in my prior review. I am pleased that the source code is provided. I assume this will be included with the publication. If so, there should be a reference to this material as an appendix or supplementary material. I am still feeling like the cardiologist expert reviewers are treated as thought they are independent from the study unless you read very deeply into the paper (page 14 of 23). The authors seem to feel this is the standard of the field and that may be the case, but it still feels disingenuous to me is all.

Also, I am still not sure how sensitive the method is to parameters that were employed (sections 4.2.1 and 4.2.4). Though the code is provided, it is so sparse and without the source images that I don't feel that I'm able to test based on this information alone. My concern is only that the parameters are very sensitive and without fine-tuning then the results will be wildly different. This is not strongly presented as a concern in the text, but it feels like a major variable for other people aiming to reproduce or benefit from this work.

Author Response

We added two files: a response to the reviewer and a comparison version, as required
